# The histone chaperone FACT modulates nucleosome structure by tethering its components

Tao Wang[1,2,*], Yang Liu[1,*], Garrett Edwards[1], Daniel Krzizike[1,2], Hataichanok Scherman[2,3], Karolin Luger[1,3,4]

**Human FAcilitates Chromatin Transcription (hFACT) is a conserved histone chaperone that was originally described as a transcription elongation factor with potential nucleosome assembly functions. Here, we show that FACT has moderate tetrasome assembly activity but facilitates H2A–H2B deposition to form hexasomes and nucleosomes. In the process, FACT tethers components of the nucleosome through interactions with H2A–H2B, resulting in a defined intermediate complex comprising FACT, a histone hexamer, and DNA. Free DNA extending from the tetrasome then competes FACT off H2A–H2B, thereby promoting hexasome and nucleosome formation. Our studies provide mechanistic insight into how FACT may stabilize partial nucleosome structures during transcription or nucleosome assembly, seemingly facilitating both nucleosome disassembly and nucleosome assembly.**

## Introduction

The organization of all genomic DNA into nucleosomes represents a formidable barrier to the cellular machinery acting on DNA. The tight wrapping of 147 bp of DNA around a histone octamer prevents access of DNA and RNA polymerases and of regulatory factors (Fischle et al, 2003; Groth et al, 2007; Kulaeva et al, 2007; Li et al, 2007; Price & D'Andrea, 2013). Therefore, the mechanism by which nucleosomes are altered dynamically during these processes is the topic of intense studies.

Nucleosomes are modular complexes whose assembly begins with the deposition of one (H3–H4)$_2$ tetramer onto ~70 bp of DNA to form a "tetrasome" (Mattiroli et al, 2017), followed by the addition of two H2A–H2B dimers that are stabilized through multiple interactions with the (H3–H4)$_2$ tetramer and flanking 2 × 35 bp of DNA (Luger & Richmond, 1998). Nucleosome disassembly likely occurs through the reverse pathway. Outside of the nucleosome, the highly basic histones are found in complex with histone chaperones that prevent their nonspecific interactions with DNA and orchestrate

their sequential deposition onto DNA (De Koning et al, 2007; Eitoku et al, 2008; Das et al, 2010).

FAcilitates Chromatin Transcription (FACT) is an abundant and essential histone chaperone that is conserved in all eukaryotes. Human FACT (hFACT) is a heterodimer composed of Spt16 and SSRP1 (Orphanides et al, 1999). Yeast FACT additionally requires Nhp6, an high mobility group box (HMGB) domain subunit that in metazoans is fused to SSRP1 (McCullough et al, 2018). The FACT complex interacts with all three RNA polymerases (Birch et al, 2009; Tessarz et al, 2014) and facilitates transcription by disrupting nucleosomes in their path and by aiding in the redeposition of histones post-transcription (Formosa et al, 2002; Belotserkovskaya et al, 2003).

Electrophoretic mobility shift assays suggest that yeast FACT binds to H2A–H2B dimer and (H3–H4)$_2$ tetramer with similar affinity (Kemble et al, 2013, 2015). Interactions between yeast FACT and H2A–H2B dimers are promoted through short acidic regions near the C-termini of each subunit that bind H2A–H2B dimer competitively and preclude H2A–H2B binding to DNA (Kemble et al, 2013, 2015). The interaction between FACT and the (H3–H4)$_2$ tetramer was confirmed in the structure of a portion of human Spt16 with an (H3–H4)$_2$ tetramer. This structure suggests that the FACT·(H3–H4)$_2$ tetramer complex is incompatible with the interactions of H3–H4 with DNA within the nucleosome (Tsunaka et al, 2016). Because H2A–H2B and H3–H4 bind distinct regions on FACT, they can bind to FACT simultaneously in the absence of DNA (Tsunaka et al, 2016).

FACT was originally described as a transcription elongation factor (Orphanides et al, 1998), but how FACT affects nucleosome structure is unknown. Using a defined chromatin template, it was shown that FACT displaces one or two H2A–H2B dimers from nucleosomes during transcription (Belotserkovskaya et al, 2003; Hsieh et al, 2013). FACT interaction with H2A–H2B is essential for this activity. Chromatin immunoprecipitation experiments in yeast suggest that yFACT also reassembles nucleosomes in the wake of RNA polymerase II (Jamai et al, 2009; Nguyen et al, 2013). Incorporation of new H3 in yeast gene bodies increases in the absence of Spt16 (Voth et al, 2014), suggesting that FACT contributes to the maintenance of preexisting tetrasomes. Thus, FACT appears to act both as a nucleosome assembly and disassembly factor. To reconcile these seemingly opposing functions,

[1]Department of Chemistry and Biochemistry, University of Colorado Boulder, Boulder, CO, USA   [2]Department of Biochemistry and Molecular Biology, Colorado State University, Fort Collins, CO, USA   [3]Institute for Genome Architecture and Function, Colorado State University, Fort Collins, CO, USA   [4]Howard Hughes Medical Institute, Chevy Chase, MD, USA

Correspondence: karolin.luger@colorado.edu
*Tao Wang and Yang Liu contributed equally to this work.

it was proposed that FACT holds the components of the nucleosome in a ternary complex during transcription elongation (Formosa, 2012). Consistent with this idea, a recent study showed that hFACT does not bind to intact nucleosomes, but only to nucleosomes with a destabilized H2A–H2B dimer (obtained by reconstitution of a nucleosome with two DNA fragments [33/112 bp] rather than with 147-bp DNA) (Tsunaka et al, 2016).

Here, we show that FACT interaction with H2A–H2B facilitates its binding to an (H3–H4)$_2$ tetramer to form a defined ternary complex. In the presence of DNA, a FACT·(H2A–H2B) complex interacts with DNA-bound (H3–H4)$_2$ tetramer, thereby tethering the components of the nucleosome. Ultimately, DNA competes FACT off the H2A–H2B dimer, resulting in the formation of hexasomes and nucleosomes.

# Results

### FACT forms a ternary complex with histones H2A–H2B and (H3–H4)$_2$

The previously reported stoichiometry of the hFACT·(H2A–H2B) complex was confirmed by sedimentation velocity analytical ultra-centrifugation (SV-AUC). The sedimentation coefficient (expressed in Svedberg units S$_{(20,W)}$) of FACT is 7.3S (Fig 1A), corresponding to a molecular weight of 198 kD and within error of the theoretical molecular weight (Table 1). Addition of equimolar amounts of H2A–H2B (which itself sediments with an S value of 1.7 to 2.5S; Fig 1B) increases the observed sedimentation to 8.3S, and the resulting apparent molecular weight is consistent with a FACT to H2A–H2B stoichiometry of 1:1 (Table 1). Doubling the amount of H2A–H2B does not further increase observed S (Fig 1A). The distribution of S values indicates a high degree of homogeneity of the FACT–histone complex.

We attempted to use the same approach to determine the stoichiometry of the FACT·(H3–H4)$_2$ tetramer complex. When 1.8 µM (H3–H4)$_2$ tetramer was mixed with an equimolar amount of FACT, visible aggregates were formed and only very low absorbance was monitored during SV-AUC. The acidic domain in Spt16 C-Terminal domain (CTD) interacts with H3–H4 nonspecifically (Tsunaka et al, 2016), and this may cause precipitation under SV-AUC conditions.

### hFACT binds to H2A–H2B and H3–H4 simultaneously

H2A–H2B binds to a conserved peptide motif in the yeast Spt16 CTD, whereas the (H3–H4)$_2$ tetramer interacts with the hSpt16 middle domain (Kemble et al, 2015; Tsunaka et al, 2016). We confirmed the simultaneous interaction of FACT with H2A–H2B and H3–H4 (shown by Tsunaka et al [2016]) by analytical ultracentrifugation equipped with a fluorescence detection system (AUC-FDS), which allows us to specifically monitor the sedimentation of fluorescently labeled constituents in complex mixtures. 100 nM Alexa 488–labeled H2A–H2B was spun in the absence or presence of 200 nM FACT. The S value of H2A–H2B increases to ~8.3S in the presence of FACT (Fig 1B and Table 1), consistent with what is seen at higher concentrations (Fig 1A). Upon addition of unlabeled H3–H4 to this complex, the sedimentation coefficient distribution indicates the presence of two species, one of which can be attributed to the FACT·(H2A–H2B) complex (~8.3S), whereas the second species (14S) likely represents

a complex of FACT, H2A–H2B, and H3–H4. To confirm this, we repeated the experiment with fluorescently labeled H3–H4 (and unlabeled H2A–H2B; FACT), and this complex sedimented at ~13.8S (Fig 1B). This demonstrates that FACT binds to H2A–H2B and H3–H4 simultaneously. The apparent molecular weight calculated from these experiments is over twice the molecular weight of a FACT·(H2A–H2B)·(H3–H4)$_2$ complex, and the stoichiometry of this assembly remains undetermined. As observed to a greater extent with unlabeled H3–H4, mixing fluorescently labeled H3–H4 (400 nM) with 200 nM FACT in the absence of H2A–H2B resulted in aggregation, as judged by the loss of >50% of fluorescence intensity (Fig S1). These results indicate that H2A–H2B dimer facilitates the proper interaction of FACT with H3–H4.

In these experiments, the H3–H4 concentration was kept below 500 nM, and thus H3–H4 exists in an equilibrium of H3–H4 dimer and (H3–H4)$_2$ tetramer (Liu et al, 2012). To resolve whether FACT·(H2A–H2B) binds to an H3–H4 dimer or an (H3–H4)$_2$ tetramer, we tested a mutated version of H3 that precludes tetramer formation (DMH3; L126A, I130A, C110E; Fig 1C) (Mattiroli et al, 2017). Available structural data show that these amino acids are not located in the FACT·(H3–H4)$_2$ interface (Tsunaka et al, 2016), and are thus unlikely to contribute to the interaction with FACT. The addition of H3–H4 DM results in a minor shift in S for the FACT·(H2A–H2B) complex (Fig 1B). The molecular weight derived from this experiment indicates a FACT·(H2A–H2B) complex, possibly with a weakly associated single H3–H4 dimer (Table 1). This indicates that H3–H4 exists as a (H3–H4)$_2$ tetramer in the complex with FACT and H2A–H2B.

### Direct interaction of H2A–H2B with H3–H4 is essential for ternary complex formation

Next, we asked whether H2A–H2B, when in complex with FACT, directly interacts with the (H3–H4)$_2$ tetramer through interactions resembling those in the nucleosome (Luger et al, 1997). To this end, we tested mutated histones H3I51A and H4Y98H that assemble into (H3–H4)$_2$ tetramers and nucleosomes in vitro (Hsieh et al, 2013) but cannot be refolded into histone octamers (Ferreira et al, 2007; Hsieh et al, 2013; Ramachandran et al, 2011) (Fig 1C). The replacement of H4Y98 with glycine in yeast is lethal (Santisteban et al, 1997). Neither side chain appears to be involved in the interaction with FACT (Tsunaka et al, 2016). AUC-FDS was repeated with tetramers refolded with mutant H3 or H4. We found that H3I51A does not significantly affect FACT interactions with a histone hexamer, whereas the more disruptive H4Y98H almost completely abolishes the integration of H3–H4 into the FACT·(H2A–H2B) complex (Fig 1D). This indicates that the interaction between the H2A docking domain and H4 is required, whereas the close packing of H3 αN with H2A and the histone fold of H3 is not (Fig 1C). Because H2A–H2B dimers and (H3–H4)$_2$ tetramers do not interact with each other at 150 mM NaCl in the absence of DNA, our results suggest that their interaction is stabilized by FACT.

### FACT neither disassembles nor interacts with intact nucleosomes in vitro

Increasing amounts of FACT were mixed with nucleosomes that had been reconstituted with a 147-bp 601 DNA fragment. Reaction products were analyzed by 5% native polyacrylamide gel electrophoresis (PAGE)

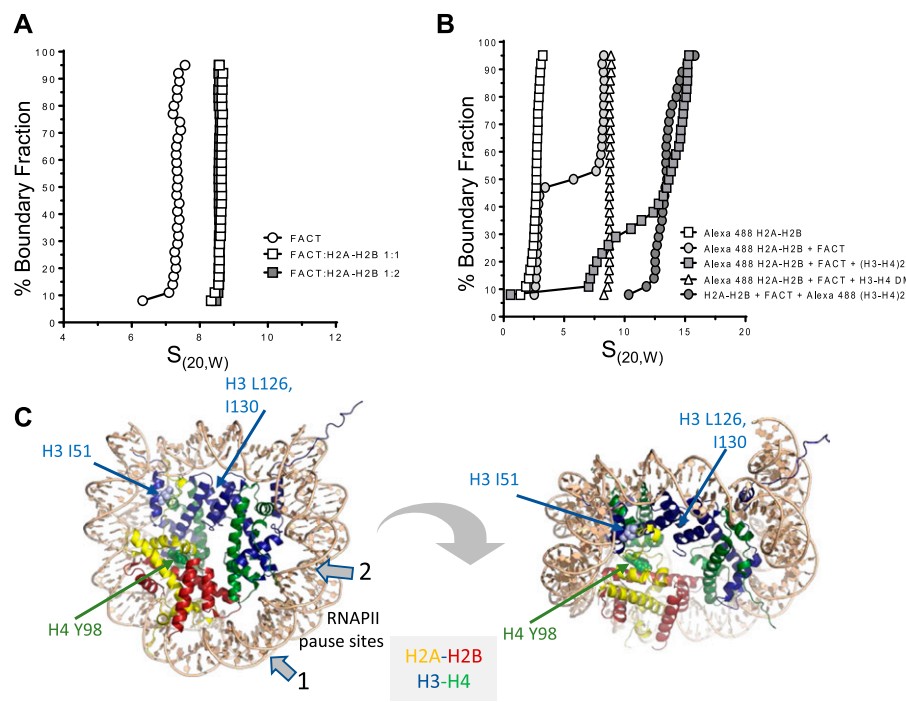

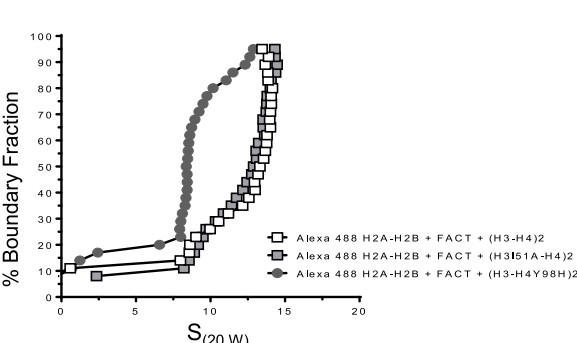

**Figure 1. H2A–H2B facilitates FACT interaction with the (H3–H4)₂ tetramer to form a ternary complex.**
**(A)** SV-AUC of histone–FACT complexes. Absorbance is represented in a van Holde–Weischet plot. 1.8 $\mu$M FACT was mixed with H2A–H2B at the indicated molar ratios. H2A–H2B sediments between 1.7 and 2.5S (B). **(B)** SV-AUC of histone–FACT complexes, by monitoring the sedimentation of fluorescently labeled histone H2B or H3 (*) with an FDS (488 nm). Open squares: 100 nM H2A–H2B*; open circles: 100 nM H2A–H2B* + 200 nM FACT; closed squares: 100 nM H2A–H2B* + 400 nM FACT + 400 nM H3–H4; closed circles: 100 nM H2A–H2B + 400 nM FACT + 400 nM H3–H4*. Open triangles: 100 nM H2–H2B*, 400 nM FACT, and 400 nM DM–H3–H4. **(C)** H3I51 and H4Y98 are shown in space filling representation in 1AOI. RNAPII pause sites specific to FACT are indicated with grey arrows; the location of H3L126 and I130 is also indicated. **(D)** SV-AUC of H2A–H2B* (100 nM) with either 400 nM H3–H4, H3I51A–H4, or H3–H4Y98H, and 400 nM FACT. Alexa 488 fluorescence was monitored with the FDS. S values from all SV-AUC experiments are listed in Table 1.

and visualized through SYBR gold staining and H2B fluorescence (Fig 2A). The intensity of the nucleosome band remained constant as FACT was titrated, indicating that under these conditions, FACT neither binds to nor disassembles nucleosomes. To exclude that this is due to the unique properties of the 601 DNA sequence, we performed the same assay with nucleosomes reconstituted with the less stable 147-bp $\alpha$-satellite DNA, with identical results (Fig S2). The inability of FACT to bind fully assembled nucleosomes was confirmed by AUC. The S value of a mononucleosome is ~11S (Yang et al, 2011) and remains unchanged upon addition of a four-fold excess of FACT (Fig 2B, Table 1). The monodisperse sedimentation coefficient distribution indicates that no significant amounts of subnucleosomal complexes were formed.

## FACT has weak tetrasome assembly function and facilitates H2A–H2B deposition onto tetrasomes and hexasomes

We next investigated whether FACT could assemble nucleosomes. To this end, an established in vitro nucleosome assembly assay was used (Mattiroli et al, 2017). First, we tested the ability of FACT to promote the first step of nucleosome assembly, that is, the deposition of an (H3–H4)₂ tetramer onto DNA to form tetrasomes. FACT was preincubated with H3–H4 before adding a 147-bp 601 DNA fragment. The reaction products were analyzed by 5% native PAGE (Fig 3A). Under these conditions (15–240 nM FACT, 30 nM H3–H4), FACT does not form perceptible aggregates with H3–H4. In the absence of FACT, only a small amount of tetrasome is formed (Fig 3A, lanes 3 and 9), whereas increasing amounts of tetrasome appeared upon titration of FACT (Fig 3A, lanes 4–8 and 10–14). However, this activity requires a large excess of FACT.

We next asked if FACT facilitates H2A–H2B addition to tetrasomes that had been preassembled from 147-bp "601" DNA and H3–H4 by salt dilution. Fluorescently labeled H2A–H2B was premixed with the indicated amounts of FACT and incubated with tetrasomes, then analyzed by native PAGE (Fig 3B). In the absence of FACT, some nucleosome and hexasome bands are observed (Fig 3B, lane 1), but the intensity of hexasomal and nucleosomal band is increased as

**Table 1.  S values and calculated molecular weights of complexes.**

| Sample | Major | Minor | Apparent molecular weight (kD) | Figure |
| --- | --- | --- | --- | --- |
| | $S_{(20,w)}$ | $S_{(20,w)}$ | | |
| FACT | 7.3 | — | Experimental = 198 | Figs 1A and 4 |
| | | | Theoretical = 203 | |
| FACT + (H2A–H2B) | 8.3 | | Experimental = 231 | Fig 1A |
| | 8.3 | 2.7 | Theoretical = 231 | Fig 1B |
| FACT + H2A–H2B + (H3–H4)$_2$ | 14.4 | | Experimental = 543 | Fig 1B (H2B*) |
| | 14.14 | | Theoretical = 220 | Fig 1B (H3*) |
| FACT + H2A–H2B + DM–H3–H4 | 8.66 | — | Experimental = 263 | Fig 1B |
| | | | Theoretical = 220 | |
| Nucleosome (147) | 11.3 | — | Experimental = 214 | Fig 2B |
| | | | Theoretical = 200 | |
| Intermed cx (79) | 12.35 | 10.7 | | Fig 5 |
| Intermed cx (87) | 12.41 | 10.7 | | Fig 5 |
| Intermed cx (93) | 12.7 | 11.3 | | Fig 5 |
| Intermed cx (95) | 12.3 | 11.6 | | Fig 5 |
| Intermed cx (99) | 12.6 | 11.3 | | Fig 5 |
| Intermed cx (147) | 11.15 | 10.15 | | Fig 5 |

FACT is titrated (Fig 3B, lanes 6–9). Again, a large excess of FACT is required for this effect, and hexasome is the main product under all conditions. This suggests that FACT promotes the first steps of nucleosome assembly but is inefficient in adding the final H2A–H2B dimer to a hexasome.

To further investigate this activity, we prepared deliberately under-assembled nucleosomes by combining optimal amounts of DNA and H3–H4 with insufficient amounts of H2A–H2B in salt reconstitution. This results in a mixture of nucleosomes, hexasomes, tetrasomes, free histones, and DNA after salt dialysis (Fig 3C, lane 3). As FACT is titrated, the fluorescence intensity from nucleosomal H4 increases, whereas the intensity of hexasomal H4 decreases (Fig 3C; quantification in Fig S3). Again, this requires very high FACT concentrations (i.e., a 64-fold excess of FACT over input DNA), suggesting that the addition of the final dimer to a hexasome is not a preferred reaction for FACT.

To test whether FACT assembles nucleosomes de novo from refolded histones and DNA, increasing amounts of FACT were mixed with a constant amount of 488-labeled H3–H4 and 647-labeled H2A–H2B before adding DNA. In the absence of FACT, histones bind nonspecifically to DNA, and very little nucleosome is formed (Fig 3D, lanes 2–4). In the presence of FACT, some nucleosomes are assembled (Fig 3D, lanes 6–10), but hexasomes are the main products, as observed before (Fig 3B). In addition, two complexes with much higher electrophoretic mobility appear in the presence of FACT (indicated by arrows, discussed below).

To confirm the quality of the nucleosomes assembled by FACT, we compared their resistance towards micrococcal nuclease (MNase) with salt-assembled nucleosomes. Free FACT and intermediate complexes containing FLAG-tagged FACT (i.e., those indicated by arrows in Fig 3D) were removed by anti-FLAG affinity purification, and the flow

through containing nucleosome assembly products was analyzed by MNase digestion (Fig S4). Nucleosomes assembled by FACT display an MNase digestion pattern that is similar to what is observed for control nucleosomes and unlike the pattern obtained from histones added to DNA in the absence of FACT (Table S1). Together, our data suggest that FACT has weak nucleosome assembly activity. This is partially through facilitating tetrasome assembly and also through aiding H2A–H2B deposition onto tetrasomes, and to a more limited extent, onto hexasomes.

### FACT·(H2–H2B) and DNA-bound H3–H4 form a ternary complex

The high molecular weight complexes observed by native PAGE during FACT-mediated nucleosome assembly assays minimally contain H4, H2B, and DNA (Fig 3D). We speculated that these complexes form as FACT deposits H2A–H2B onto tetrasomes. To test this, we reconstituted tetrasome with 147-bp DNA and H3–H4 by salt dilution, and this was added to FACT premixed with H2A–H2B (Fig 4A). FACT does not bind to tetrasome in the absence of H2A–H2B (Fig 4A, left panel, lane 4). However, when prebound with H2A–H2B, FACT binds to tetrasomes efficiently and forms two intermediate complexes (Fig 4A, lanes 5–6), as observed in Fig 3D. To confirm that the complexes contain all histones, DNA and FACT, we analyzed these samples by immobilizing FLAG-tagged FACT and associated proteins on an M2 affinity column and visualized DNA, H4, and H2B fluorescence of bound and unbound fractions (Fig 4B). Nucleosomes (yellow), subnucleosomal complexes (green and red), and free DNA were enriched in the flow through (lane 3), whereas only the intermediate complexes containing FLAG-tagged FACT are specifically eluted with the FLAG peptide. This complex also contains DNA, H2B, and H4 (yellow, lane 4).

**A**

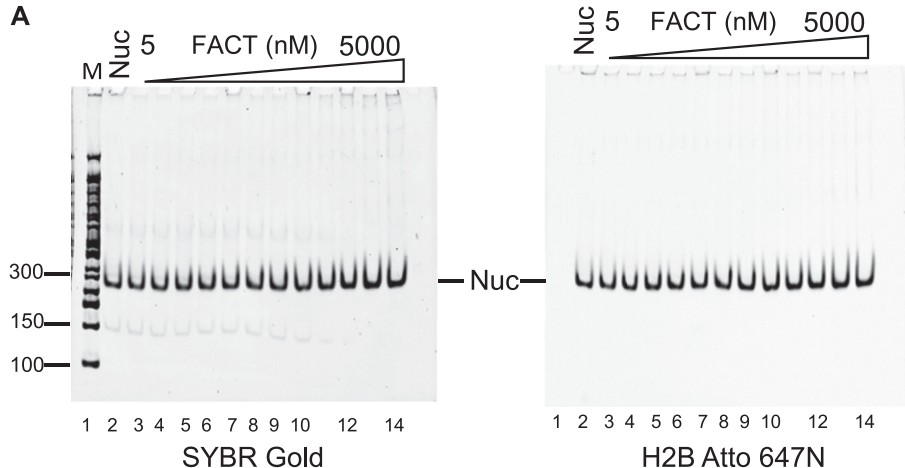

Figure 2. **FACT neither disassembles nor binds properly assembled nucleosomes.**
**(A)** 10 nM nucleosome, reconstituted with labeled H2B on 146 bp of DNA, was incubated with increasing amounts of FACT (10 nM–5 $\mu$M) and analyzed by native PAGE. The gel was visualized by SYBR Gold for DNA (left panel), and by H2B fluorescence (right panel). **(B)** The sedimentation behavior of nucleosomes remains unchanged upon addition of FACT. 130 nM nucleosomes were mixed with 520 nM FACT, and SV-AUC was performed by monitoring absorbance from DNA. Addition of FACT does not increase the S value of the nucleosome.

**B**

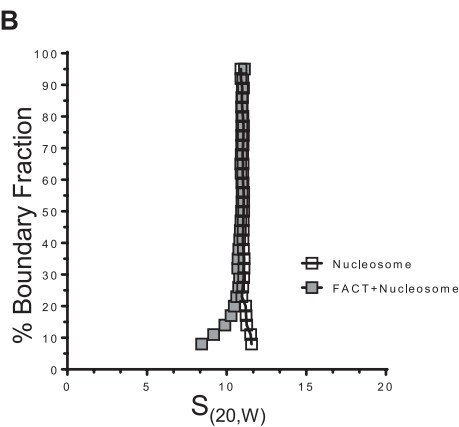

Only the faster migrating intermediate complex is formed at 60 nM H2A–H2B (Fig 4A, left panel, lanes 5–6). At higher H2A–H2B concentrations (240 nM), most of the material was in the slower migrating species (Fig 4C, lane 4). This indicates that the slower migrating complex contains one more H2A–H2B than the faster migrating complex. Importantly, with 147-bp DNA, either intermediate complex only formed at >20-fold excess of FACT and H2A–H2B over (H3–H4)$_2$ tetrasome.

### DNA displaces bound FACT from the complex and facilitates H2A–H2B deposition to assemble nucleosomes

To test whether DNA beyond the ~80 bp organized by the (H3–H4)$_2$ tetramer contributes to the intermediate complex, we attempted to assemble it onto 79-bp DNA. The (H3–H4)$_2$ tetrasome was first reconstituted with 79-bp DNA fragment and then combined with FACT·(H2A–H2B) (Fig 5A). As shown before with longer DNA, FACT does not bind to the (H3–H4)$_2$ tetrasome in the absence of H2A–H2B dimer (lane 3). Similarly, H2A–H2B does not bind properly to (H3–H4)$_2$ tetrasome in the absence of FACT but rather forms aggregates that do not enter the gel (lane 2). Thus, with 79-bp DNA, the intermediate complexes form only with tetrasomes and FACT·(H2A–H2B), as observed previously with 147-bp DNA. However, no nucleosomes are formed, and unlike 147-bp DNA which only

assembles into intermediate complexes at very high FACT and H2A–H2B concentrations, the intermediate complex with 79-bp DNA was observed at equimolar FACT to (H3–H4)$_2$ to H2A–H2B ratios.

To better define the effect of DNA length in intermediate complex formation and nucleosome assembly, DNA of different lengths (79, 87, 93, 95, 99, and 147 bp) were tested. FACT, (H3–H4)$_2$ tetrasome, and H2A–H2B were mixed at a 1:1:2 ratio. Only shorter DNA (79, 87, and 93 bp) form near-homogenous intermediate complexes (Fig 5B, lanes 3 and 5). With longer DNA (95 and 99 bp), the intermediate complex still forms, but the main products are partially assembled nucleosomes that contain H2A–H2B, H3–H4, and DNA but no FACT. When the DNA is 147-bp long, almost no intermediate complex is formed, and the assembly product is mostly nucleosome (Fig 5B).

We analyzed these reactions by AUC-FDS at 488 nm, where only complexes containing fluorescently labeled H4 are visible (Fig 5C; S values listed in Table 1). The intermediate complexes on either 79-bp (12.35S) or 87-bp (12.41S) DNA are quite homogenous in size, and only small amounts of a minor species with a lower S value was observed. For complexes formed with 93-, 95-, and 99-bp DNA, the major species are also likely the FACT-bound intermediate complex, whereas the other components probably represent nucleosomes or their assembly intermediates in the absence of FACT. For the intermediate complexes with 147-bp DNA, the major species sediments as a nucleosome, with some hexasome and tetrasome, and

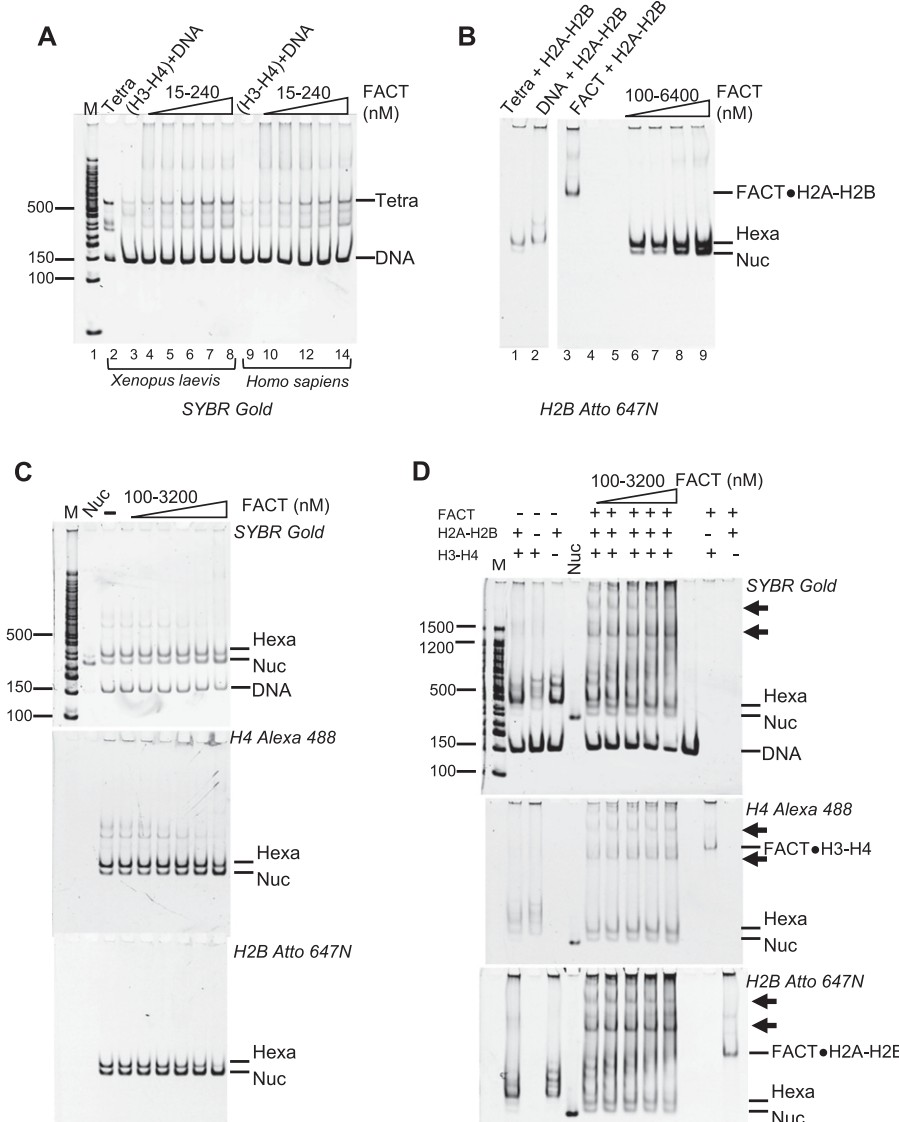

**Figure 3. FACT has moderate nucleosome assembly activity.**
**(A)** FACT has moderate tetrasome assembly activity. 147-bp 601 DNA (30 nM) was incubated with FACT (15–240 nM) pre-equilibrated with 30 nM *X. laevis* or *H. sapiens* $(H3–H4)_2$, separated by PAGE, and visualized by SYBR Gold (DNA). Lane 2 shows a salt-assembled tetrasome on the same DNA (Tetra). **(B)** FACT facilitates the deposition of H2A–H2B dimers onto the tetrasome, resulting in hexasome and nucleosome assembly. 60 nM salt-reconstituted tetrasome (lane 2) and 30 nM Atto 647N–labeled H2A–H2B were incubated with increasing amounts of FACT (10–1,280 nM), and gels were visualized by H2B fluorescence. **(C)** FACT facilitates the deposition of H2A–H2B onto the hexasome. 20 nM under-assembled nucleosome with fluorescence labels on H4 (Alexa 488) and H2B (Atto 647N) (lane 3) was incubated with increasing amounts of FACT, and the gel was visualized as indicated. **(D)** FACT (100–3,200 nM) promotes hexasome and nucleosome assembly from free histones (50 nM H3–H4; 100 nM H2A–H2B) and DNA (25 nM). FACT (at the indicated concentrations) was incubated with histones at RT for 10 min and then DNA was added, and products were visualized as indicated. The two intermediate complexes are indicated by arrows.

little to no intermediate complex, consistent with what was observed by native PAGE (Fig 5B). Together, these experiments demonstrate that when free DNA extends from the tetrasome, it can compete FACT from H2A–H2B and dislodge it from the complex. This effectively results in H2A–H2B deposition onto $(H3–H4)_2$ tetrasomes to form hexasomes, and ultimately nucleosomes. Thus, the intermediate complex is quite stable when the DNA is of limited length and unable to compete with FACT. In contrast, at least a 20-fold molar excess of FACT over $(H3–H4)_2$ tetrasome is needed to "win" the competition with DNA and to form the intermediate complex with 147-bp DNA. This is consistent with the published FACT·(H2A–H2B) crystal structure which shows that FACT·(H2A–H2B) interaction is incompatible with DNA·(H2A–H2B) interaction (Hsieh et al, 2013; Kemble et al, 2015). Importantly, our data demonstrate that we indeed observe intermediate states of FACT in the act of assembling nucleosomes or stabilizing partially disassembled nucleosomes.

## H2A–H2B interacts with both FACT and H3–H4 in the intermediate complex

Having established that H2A–H2B is required for the intermediate complex, we tested whether direct interactions between H2A–H2B and FACT are required. FACT ΔCTD ($Spt16_{1–934}$) was previously shown to be deficient for H2A–H2B binding (Winkler et al, 2011). AUC-FDS confirmed that this complex does not bind to H2A–H2B because no change in S value was observed upon mixing 100 nM Alexa 488–labeled H2A–H2B with 200 nM FACT ΔCTD (Fig S5) and no intermediate complex was formed on a 79-bp DNA fragment (Fig 6A, lane 5). Instead, aggregates are formed similar to those obtained in the absence of FACT.

Next, we queried the requirement for direct H2A–H2B and H3–H4 interactions by testing mutant histones H3I51A and H4Y98H (Fig 1C) on a 79-bp DNA fragment. Neither mutation appears to significantly

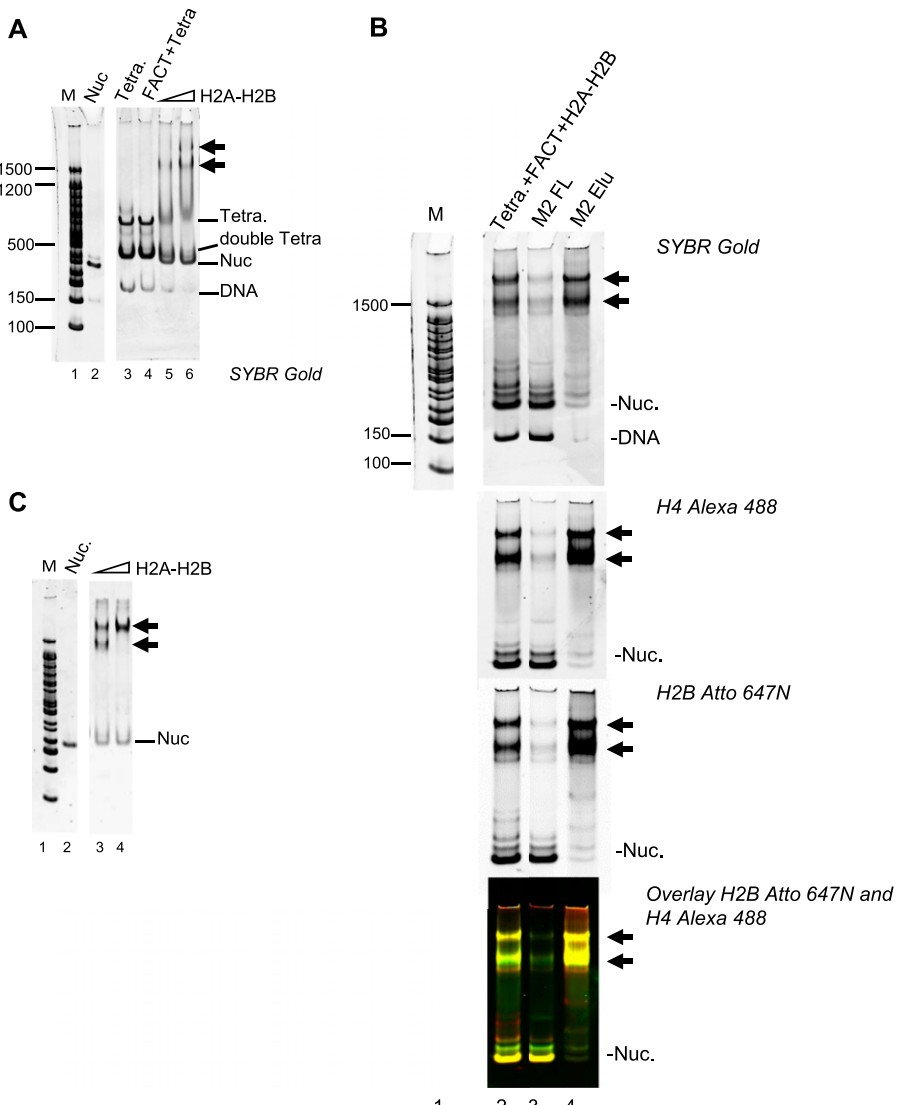

**Figure 4. H2A–H2B promotes FACT interaction with the tetrasome.**
**(A)** FACT (800 nM) was mixed with 60 nM or 120 nM H2A–H2B before addition of tetrasome (40 nM) assembled with 147-bp DNA. Double tetra: two $(H3–H4)_2$ assembled on 147-bp DNA. **(B)** FACT (1.3 $\mu M$) was premixed with 650 nM H2A–H2B (labeled with Atto 647), and 150 nM tetrasome (labeled with Alexa 488) was added. Intermediate complexes (indicated by arrows) were enriched over M2 resin and analyzed by 5% PAGE. The marker (lane 1) is on the same gel, but intervening lanes were removed. **(C)** To determine if the two intermediate complexes (indicated with arrows) contain the same relative amount of H2A–H2B, FACT was mixed with an even larger excess of H2A–H2B (160 and 240 nM) before adding tetrasome.

impede the formation of intermediate complex (Fig 6B). This indicates that the H2A–H2B·(H3–H4) in the intermediate complex is further stabilized by DNA. This result is consistent with our previous findings that H3I51A and H4Y98H cannot be refolded into histone octamers in the absence of DNA, but still assemble into $(H3–H4)_2$ nucleosomes in the presence of DNA in vitro (Hsieh et al, 2013). This indicates a role for DNA in tethering H3–H4 onto the FACT·(H2A–H2B) complex.

## FACT facilitates transcription by RNA polymerase II

Finally, we wanted to reinvestigate the effect of FACT on transcription by using a recombinant minimal in vitro transcription system. In previous reports, yeast FACT (in absence of Nhp6) was shown to have only a moderate effect on the amount of full-length RNAPII transcript when a remodeling factor was present (Kuryan et al, 2012). In light of our finding that human FACT binds to hexasomes, we wanted to test if it affects yeast RNAPII pause sites. To

this end, nucleosomes were reconstituted on the templates shown in Fig 7A. Nucleosomes were kept at 5 nM, and FACT was added at the indicated amounts. Transcription was initiated by the addition of NTP, and after a 15-min incubation, RNA transcripts were analyzed on a sequencing gel (Fig 7B).

As previously described by Kuryan et al (2012), nucleosomes preclude formation of full-length transcript (compare Fig 7C, lanes 2–3), and this repression was relieved upon addition of the ATP-dependent chromatin remodeler remodelling the structure of chromatin (RSC) and the histone chaperone Nap1 (lane 5). FACT on its own has only moderate effects on the amount of full-length transcript but appears to promote entry into the nucleosome as demonstrated by the disappearance of the shortest transcripts.

Notably, two intermediate-length transcripts indicative of polymerase pause sites were observed in the presence of FACT, one at ~85 bp, or 35 bp into the nucleosome, and one at ~105 bp, or 55 bp into the nucleosome (Fig 7C, lanes 6–10 and 1C). The first site coincides with the interaction of DNA with the H2B L1 loop, and is thus

# A

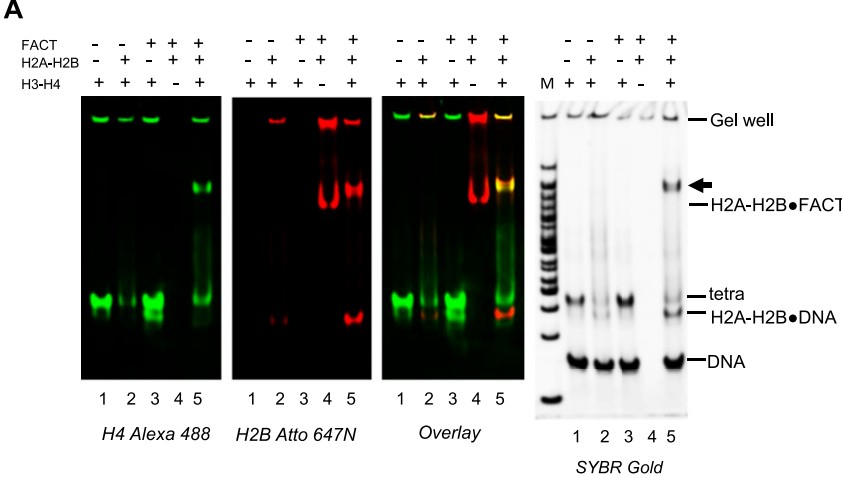

1 2 3 4 5 — *H4 Alexa 488* · 1 2 3 4 5 — *H2B Atto 647N* · 1 2 3 4 5 — *Overlay* · M 1 2 3 4 5 — *SYBR Gold*

**Figure 5. DNA competes FACT away from H2A–H2B and facilitates formation of hexasomes/nucleosomes.**

**(A)** The intermediate complex is also formed with 79-bp DNA. FACT was preincubated with H2A–H2B for 10 min at room temperature and tetrasome (pre-assembled with (H3–H4)$_2$ and 79-bp DNA) was then incubated for 30 min at room temperature. In the final reaction, all components are at ~400 nM. The 5% native gel was visualized by SYBR Gold staining, or through H4 (Alexa 488) or H2B (Atto 647). **(B)** DNA competes FACT off H2A–H2B and facilitates H2A–H2B deposition onto tetrasomes. FACT was premixed with H2A–H2B, and tetrasome with different DNA length (79–147 bp) was added. In the final reaction, Alexa 488–labeled tetrasome concentration is ~500 nM. FACT is at 500 nM, and Atto 647N–labeled H2A–H2B is ~1,000 nM. The 5% native gel was visualized by H4 (Alexa 488; green) or H2B (Atto 647; red). Top panel: an overlay of scans at both wavelengths; bottom panel: SYBR Gold. **(C)** Complexes formed with different length DNA fragments, as in **(B)**, were analyzed by SV-AUC (FDS at 488 nm) to monitor fluorescently labeled H4. S values are listed in Table 1.

# B

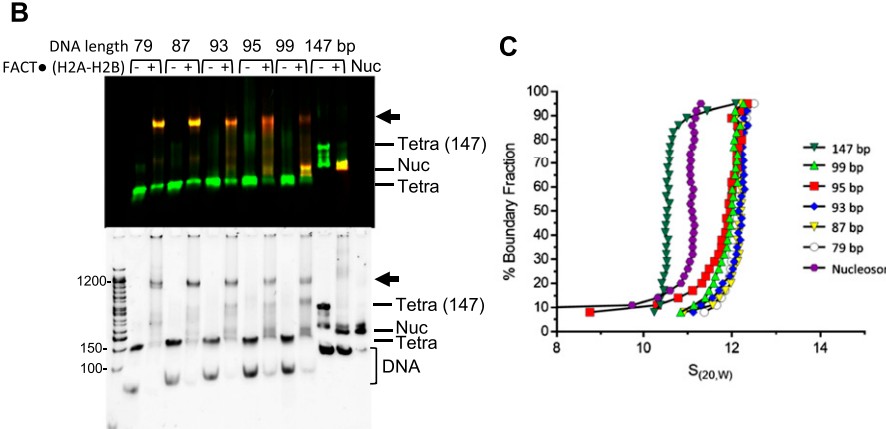

# C

consistent with polymerase passage through DNA without being able to progress through the H4–H2B four-helix bundle. The second pause site coincides with the H3–H4 α1α1 interface and could stem from the interaction of FACT with a hexasome (i.e., the intermediate complex described above). In the presence of RSC, FACT stimulates transcription in a dose-dependent manner; the decrease in full-length transcript at high FACT concentration had been observed previously (Belotserkovskaya et al, 2003). In the presence of RSC and FACT, no polymerase pausing is observed.

## Discussion

The mechanism by which FACT, an essential and conserved histone chaperone complex, facilitates the processes of gene transcription, DNA replication, and DNA repair in the context of chromatin is unknown (Wittmeyer & Formosa, 1997; Schlesinger & Formosa, 2000; Keller & Lu, 2002). Our results with purified FACT and nucleosomes/histones provide mechanistic insight into how hFACT stabilizes intermediate states of the nucleosome, seemingly promoting both nucleosome disassembly and assembly, and ultimately regulating access to nucleosomal DNA.

FACT alone does not bind to or disassemble fully assembled nucleosomes, consistent with recent data (Tsunaka et al, 2016), but

contradicting published work with HeLa nucleosomes in pull-down assays (Belotserkovskaya et al, 2003). HeLa nucleosomes carry an abundance of posttranslational modifications and likely exist in a mixture of intact and partially destabilized nucleosomes, which are further destabilized with an excess of FACT. In our own published work using fluorescence resonance energy transfer-based assays, we have determined that FACT interacts with nucleosomes (Winkler et al, 2011). The observed fluorescence resonance energy transfer changes might be attributed to interactions of FACT with free labeled H2A–H2B dimer, or with hexasomes that might exist at these very low nucleosome concentrations. Analytical ultracentrifugation experiments provide first-principle information on the behavior of macromolecules in solution, and allowed us to determine the inability of FACT to bind to intact nucleosomes. FACT is also unable to disassemble nucleosomes, and likely requires destabilizing activities such as RNAPII or ATP-dependent chromatin remodeling factors for its action. In addition, the presence of HMGB DNA-binding protein cofactors is required for FACT to reorganize nucleosomes (McCullough et al, 2018). Overall, the consensus is that FACT per se lacks the ability to modulate nucleosome.

FACT is particularly relevant during gene transcription, where the requirement for de novo nucleosome assembly is not as stringent as during replication or DNA repair. Especially at slower transcription rates, tetrasomes or hexasomes survive the passage of

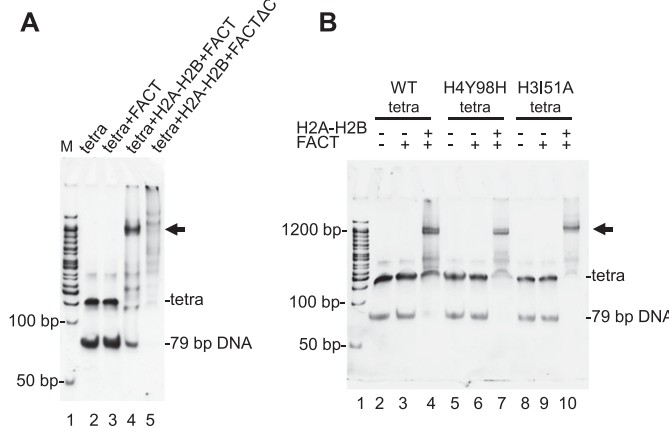

**Figure 6. FACT·(H2A–H2B) interactions are required for the intermediate complex.**
**(A)** 500 nM FACT or FACT ΔC was premixed with 1,000 nM H2A–H2B, then tetrasome with 79-bp DNA (500 nM) was added. Intermediate complex is indicated by an arrow. **(B)** FACT·(H2A–H2B) was incubated with WT or mutant (Y98H or I51A) tetrasome, reconstituted with 79-bp DNA. Concentrations were as in **(A)**.

RNAPII (Thiriet & Hayes, 2006; Dion et al, 2007; Jamai et al, 2007). Indeed, we found that FACT is only moderately effective in mediating the deposition of (H3–H4)$_2$ tetramer onto DNA. FACT can deposit an H2A–H2B dimer onto a tetramer to form a hexasome, whereas the addition of the second H2A–H2B dimer is not as favorable and requires a large excess of FACT.

FACT, when bound to a histone H2A–H2B dimer, assembles into a stable intermediate complex with (H3–H4)$_2$ bound to DNA (a tetrasome). This complex differs from the complex with histones in the absence of DNA described previously (Tsunaka et al, 2016) and confirmed here. Because 79 bp of DNA, the minimal length to wrap the entire (H3–H4)$_2$ tetramer, is sufficient to form this complex, a significant portion of the 147-bp DNA that would be tightly bound in the context of a nucleosome is freely accessible. We suggest a model where free DNA (when not otherwise engaged with a transcribing RNAPII or bound by transcriptional regulators) can compete with FACT for H2A–H2B, resulting in FACT displacement and the formation of a hexasome, which appears to be the predominant end product of FACT-mediated nucleosome assembly in vitro (Fig 8). This explains why the intermediate complex is observed on longer DNA fragments only when a large excess of FACT is used, and implies that the interaction of the FACT·(H2A–H2B) complex with a hexasome to potentially deliver the second dimer is unfavorable. A stable complex between FACT and a hexasome, in which only ~80 bp of DNA are tightly bound, is also consistent with the idea that yeast FACT "reorganizes nucleosomes" (Xin et al, 2009). A similar complex was observed with a nucleosome that had been reconstituted with two DNA fragments (33 and 112 bp), which significantly destabilizes the nucleosome and results in the loss of one H2A–H2B dimer and possibly the short DNA fragment (Tsunaka et al, 2016).

H2A–H2B does not interact with H3–H4 in the absence of DNA at physiological conditions, but is able to do so when prebound to FACT. The interactions between these histones in the FACT complex are similar to the nucleosome. FACT CTD interaction with H2A–H2B dimer is required to form a complex with a hexasome. The contributions of other functional domains of FACT are still unknown,

but some are likely to also participate. For example, recent studies with an artificially tethered HMGB domain highlight its importance for yeast FACT function (McCullough et al, 2018).

Our results suggest a mechanism by which FACT facilitates gene transcription through a nucleosome. We find that FACT, in conjunction with RSC, stimulates the formation of a full-length transcript, unlike what was observed previously using a similar transcription system (Kuryan et al, 2012). One plausible explanation for this discrepancy is that the published study used yeast FACT in the absence of the HMGB domain protein Nhp6; in the human FACT complex used here, the HMGB domain subunit is fused to SSRP1 and this was shown to be important for nucleosome reorganization (McCullough et al, 2018).

Shorter transcripts resulting from RNAPII pausing are observed in the presence of FACT, independent of the chromatin remodeling factor RSC. Because the position of the nucleosome on 601 DNA is known with high accuracy, these pause sites give insight into where FACT might interact with nucleosomal DNA during transcription (Fig 1C). The concept that FACT tethers partially assembled nucleosomes is consistent with the finding that histone turnover rates are enhanced in yeast when FACT activity is compromised (Jamai et al, 2009). During gene transcription, H2A–H2B is much more dynamic than H3–H4 (Jamai et al, 2007). FACT might capture an H2A–H2B dimer that had been displaced from the nucleosome by the advancing RNA polymerase, and reinstate it on the tetrasome by forming a ternary complex such as the one we describe here. In this model, FACT is displaced by DNA when that DNA is no longer engaged with the polymerase. FACT or another H2A–H2B chaperone could then facilitate deposition of the second H2A–H2B onto a hexasome (without forming a ternary complex) to complete nucleosome reassembly. Similarly, FACT could temporarily stabilize a partial nucleosome complex consisting of one H2A–H2B dimer (whose DNA is engaged with the polymerase) and one DNA-bound (H3–H4)$_2$ tetramer. Our model explains the dose–response curve of FACT on transcript lengths, because at very high FACT concentrations, it can no longer be displaced and inhibits transcription.

## Materials and Methods

### Reagents

Recombinant histone genes (sequences derived from either *Xenopus laevis* or *Homo sapiens*) were expressed and purified as described previously (Luger et al, 1999). H4T71C mutant was used for labeling of H3–H4 with Alexa 488, whereas H2B T112C was used to label H2A–H2B with Atto 647N. 147-bp or 207-bp 601 DNA was prepared as described, and nucleosomes with 147-bp or 207-bp DNA were reconstituted by salt dialysis (Dyer et al, 2004).

### FACT expression and purification

The purification of human FACT (full length or FACT with Spt16$_{1–934}$; ΔCTD) was adapted from published work with minor changes (Winkler et al, 2011). FACT complexes were purified over a 5-ml prepacked HisTrap HP column, followed by purification over a 5-ml

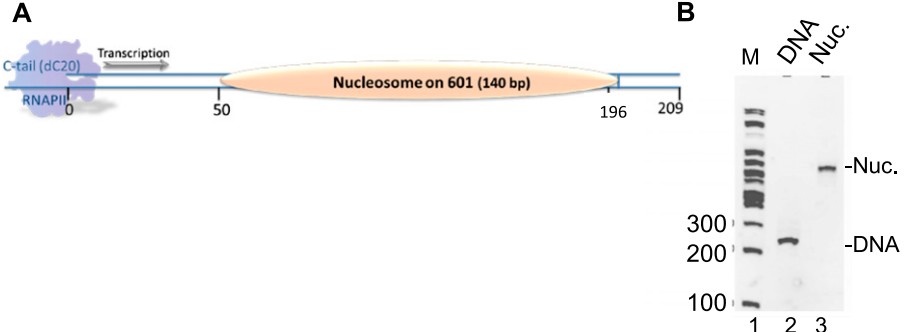

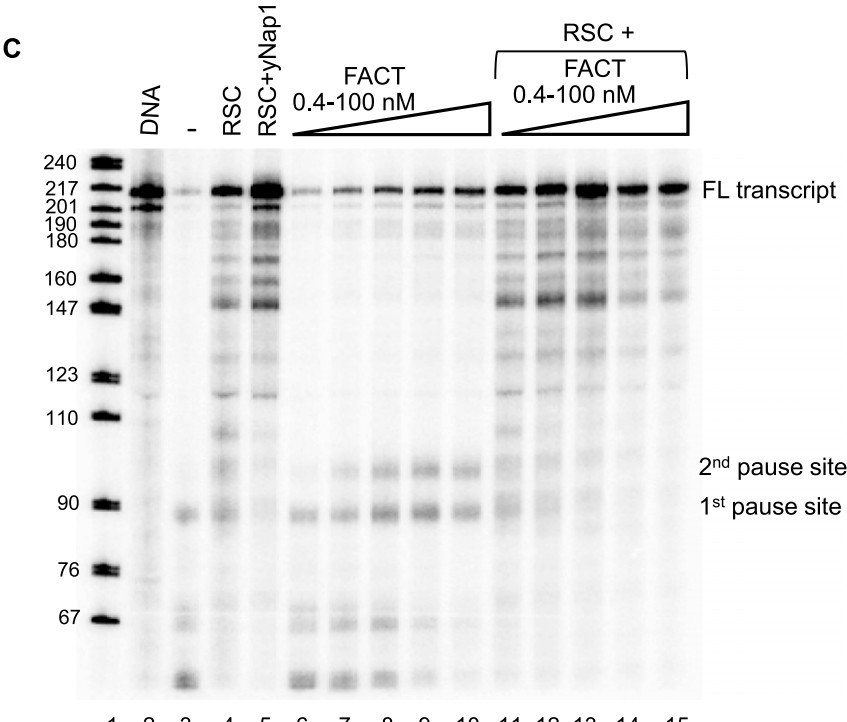

**Figure 7.  FACT affects RNAPII-mediated transcription through a nucleosome.**
**(A)** Schematic of the transcription template. **(B)** Nucleosomes were reconstituted on the transcription template shown in **(A)** and analyzed by native PAGE. Note the absence of free DNA. **(C)** The transcription output was analyzed on a 6.5% acrylamide sequencing gel and visualized by radioactivity (P32). Pause sites are also indicated in Fig 1C.

prepacked HiTrap Q HP column. The final step was a Superdex 200 10/300 size exclusion column in 300 mM NaCl, 20 mM Tris, pH 8.0, 5% glycerol, 0.01% CHAPS, 0.01% octyl glucoside, and 1 mM TCEP. FACT was stored in 150 mM NaCl, 20 mM Tris, pH 8.0, 10% glycerol, 0.01% CHAPS, 0.01% octyl glucoside, and 1 mM TCEP (buffer A). CHAPS and octyl glucoside help avoid nonspecific protein–protein interaction. All columns were purchased from GE Healthcare.

### SV-AUC

To determine the stoichiometry of the FACT·(H2A–H2B) complex, SV-AUC with absorbance optics was performed in buffer containing 150 mM NaCl, 20 mM Tris pH 8.0, and 1 mM TCEP. 1.8 $\mu$M FACT was incubated with the indicated amount of H2A–H2B at room temperature for 10 min, then spun at 30–35,000 rpm and 20°C in a Beckman XL-A ultracentrifuge, using an An60Ti rotor. Partial

specific volumes of the samples were determined using UltraScan 3 version 2.0 (Department of Biochemistry, The University of Texas Health Science Center at San Antonio). Time invariant and radially invariant noise was subtracted by 2-dimensional-spectrum analysis followed by genetic algorithm refinement and Monte Carlo analysis to resolve the sedimentation coefficients (s, converted to Svedberg units [S] by multiplying by $10^{-13}$) and frictional ratios ($f/f_0$) of significant species in each sample, from which the apparent molecular weights were extracted (Cao & Demeler, 2008; Brookes et al, 2010). Sedimentation coefficient distributions G(s) were obtained with enhanced van Holde–Weischet analysis, plotted with sedimentation coefficients (s) converted to Svedbergs (Demeler & van Holde, 2004). Analyses were performed using Ultrascan 3 version 2.0 and distributions were plotted using Graphpad Prism.

To determine if FACT binds nucleosomes, 150 nM nucleosome was incubated with 600 nM FACT at room temperature for 10 min,

**Life Science Alliance**

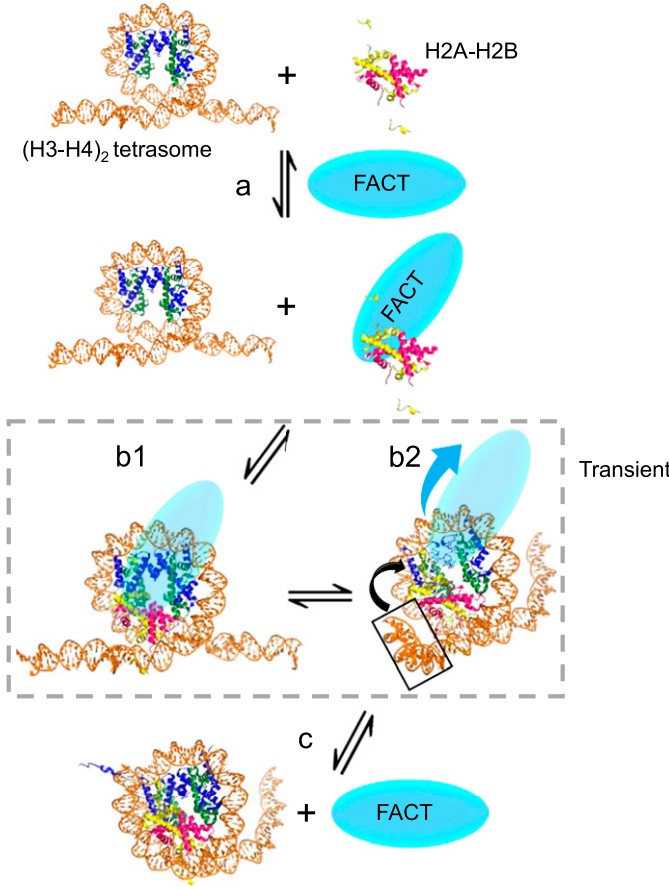

**Figure 8. Model of FACT action in nucleosome/hexasome assembly.**
Step a: FACT binds to H2A–H2B dimer; step b1: FACT chaperones H2A–H2B onto (H3–H4)$_2$ tetrasome; step b2: free DNA end competes FACT off H2A–H2B; step c: hexasome is formed.

then spun using an An50Ti or An60Ti rotor (Beckman Coulter) at 32,000 rpm, 20°C in a Beckman XL-A ultracentrifuge using absorbance optics. Data analysis was performed as described above.

To determine if FACT binds a histone hexamer, SV-AUC experiments were performed in a Beckman XL-A ultracentrifuge equipped with an Aviv FDS, using an An60Ti rotor (Beckman Coulter) with standard epon 2-channel centerpiece cells. Alexa 488–labeled *X. laevis* histones (maleimide linked at either H4 T71C or H2B T112C) were fixed at either 100 or 200 nM and incubated with FACT. Sedimentation was then monitored with fluorescence optics (excitation 488 nm, emission >505 nm) at 20°C using speeds of 40 or 45,000 rpm.

To determine the effect of DNA length on FACT·(H2A–H2B)·(H3–H4)$_2$ complex formation, SV-AUC experiments were performed in a Beckman XL-A ultracentrifuge with Aviv FDS, using an An50Ti rotor with standard epon 2-channel centerpiece cells. *H. sapiens* histones were labeled at H4 T71C via maleimide linkage with Alexa 488 and were reconstituted with H3 and DNA with the indicated lengths (79, 87, 93, 95, 99, or 147 bp) to form (H3–H4)$_2$ tetrasomes by salt reconstitution. Labeled tetrasomes were adjusted to 350 nM (in buffer containing 20 mM Tris HCl, pH 7.5, 150 mM NaCl, and 1 mM EDTA), and combined with equimolar amounts of

preincubated FACT·(H2A–H2B). Sedimentation was then monitored using fluorescence optics at 20°C and 30,000 rpm. Ultrascan 3 version 4.0 was used for all analysis, and distributions plotted using Graphpad Prism.

All data analyses were performed on the UltraScan LIMS cluster at the Bioinformatics Core Facility at the University of Texas Health Science Center at San Antonio and the Lonestar cluster at the Texas Advanced Computing Center supported by National Science Foundation TeraGrid grant #MCB070038.

### Nucleosome assembly/disassembly assay

To test if FACT disassembles nucleosomes, FACT (5 nM–5 µM) was titrated into nucleosomes previously assembled onto 147-bp "601" DNA (10 nM) and incubated in buffer A at room temperature for 1 h, then analyzed by 5% PAGE, and run at 150 V, 4°C for 60 min.

To determine if FACT facilitates tetrasome assembly, FACT (15–240 nM) was titrated into refolded (H3–H4)$_2$ tetramer (30 nM) and incubated in buffer A at room temperature for 10 min. DNA (30 nM) was added, and tetrasome formation was analyzed by native PAGE as described above.

To determine if FACT facilitates H2A–H2B deposition onto tetrasome, Atto 647N–labeled H2A–H2B dimer (30 nM) was mixed with varying amounts of FACT and incubated at room temperature for 10 min. 60 nM tetrasome was added and incubated at room temperature for another 30 min. The reaction was performed in buffer A and analyzed as described above.

To test whether FACT facilitates H2A–H2B deposition onto hexasomes, under-assembled nucleosomes were reconstituted using an optimal ratio of H3–H4 to DNA and subsaturating amounts of H2A–H2B. FACT (10–1,280 nM) was then titrated into 20 nM under-assembled nucleosome and analyzed as described above.

To investigate if FACT facilitates nucleosome assembly, Alexa 488–labeled H3–H4 (500 nM) and Atto 647N–labeled H2A–H2B dimer (500 nM) were mixed with or without FACT (1 µM) and incubated at room temperature for 10 min. 207-bp 601 DNA (250 nM) was added, incubated at room temperature for 30 min, and analyzed by native PAGE.

To determine if FACT assembles intermediate complexes with tetrasome with 147-bp DNA, 850 nM FACT was mixed with increasing amounts of H2A–H2B (160 and 240 nM) and then 40 nM tetrasome was added.

### MNase digestion assay

3.2 µM FACT was mixed with 400 nM H3–H4 and 400 nM H2A–H2B dimer, and incubated at room temperature for 10 min. 100 nM 147-bp DNA was then added and incubated at room temperature for 30 min in buffer A. Increasing amounts of MNase (100 U, 200 U, and 400 U) was added to the anti-FLAG purified supershifted complexes and incubated at 37°C for 10 min. The reaction was quenched by adding 5 µl 0.5 M EDTA. 621-bp DNA was added as a recovery standard. DNA fragments were purified through a Mini-iElute PCR Purification kit (QIAGEN). DNA fragments were quantified by 2100 Bioanalyzer (Agilent) and analyzed by 2100 Expert software. To determine the stability of nucleosome assembled by FACT, MNase

digestions for FACT-assembled nucleosomes were performed and analyzed as described above.

### In vitro transcription assay

The in vitro transcription assay was adapted from published work (Kuryan et al, 2012). 141-bp 601 DNA, flanked by 50-bp polylinker DNA and 20-bp plasmid DNA were joined to a single-stranded C tail as a binding site for RNAPII (sequence given below). Mono-nucleosomes were reconstituted by salt dialysis; special care was taken to reach a correct assembly ratio without free DNA. All transcription reactions were performed in 25 mM Hepes, pH 7.5, 10 mM $MgCl_2$, 2.5 mM KCl, 10% glycerol, 1 mM DTT, and 250 ng/$\mu$l BSA. Nucleosome concentration was kept at 0.5 nM. FACT was titrated from 0.4 to 100 nM, and transcription was initiated by adding dNTP. RNA transcripts were analyzed on a 6.5% acrylamide sequencing gel. RSC and RNA pol II were prepared as previously described (Izban & Luse, 1991).

### C-tail sequence

CCCCCCCCCCCCCCCCCCCCCCTGTGGGCCCTTCTTTTTCGTTTGGCGTCTCTAG ACACCCGGGAAGAAAAAGCAAACCGCAGA.

### Ligated to 180-bp 601 sequence (only bottom strand shown, BamH1 and EcoRV sites underlined)

GATCCATGCACA**GGATGTA**TATATCTGACACGTGCCTGGAGACTAGGGAGTA-ATCCCCTTGGCGGTTAAAACGCGGGGGACAGCGCGTACGTGCGTTTAAGCG-GTGCTAGAGCTGTCTACGACCAATTGAGCGGCCTCGGCACCGGGAT**TCTCCA**-GGAATTCAAGCTTCCCGGGGGGGAT.

## Supplementary Information

## Acknowledgements

We thank Dr. Serge Bergeron for valuable suggestions regarding the nucleosome assembly assay. This study was supported by the Howard Hughes Medical Institute and by NIH-GM-067777.

## Author Contributions

T Wang: conceptualization, formal analysis, validation, investigation, methodology, and writing–original draft.
Y Liu: conceptualization, formal analysis, validation, investigation, visualization, methodology, and writing—review and editing.
G Edwards: investigation, methodology, and writing—review and editing.
D Krzizike: methodology.
H Scherman: investigation and methodology.
K Luger: conceptualization, resources, formal analysis, supervision, funding acquisition, investigation, project administration, and writing—review and editing.

## Conflict of Interest Statement

The authors declare that they have no conflict of interest.

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
