## [Reviewer comments · Life Science Alliance]

The histone chaperone FACT modulates nucleosome structure by tethering its components

Tao Wang¹, Yang Liu, Garrett Edwards, Daniel Krzizike, Hataichanok Scherman, and Karolin Luger

DOI: 10.26508/lsa.201700107

Review timeline:

Submission date:	14 June 2018
Editorial Decision:	14 June 2018
Revision received:	26 June 2018
Editorial Decision:	27 June 2018
Accepted:	29 June 2018

Report:

(Note: Letters and reports are not edited. The original formatting of letters and referee reports may not be reflected in this compilation.)

Please note that the manuscript was previously reviewed at another journal and the reports were taken into account in inviting a revision for publication at *Life Science Alliance* prior to submission to *Life Science Alliance*.

1st Editorial Decision

14 June 2018

Thank you for transferring your manuscript entitled "The histone chaperone FACT modulates nucleosome structure by tethering its components" to Life Science Alliance. The manuscript was assessed by expert reviewers at another journal before, and the editor confidentially transferred these reports to us with your permission.

The reviewers noted some redundancies with prior work, but this is not a concern for publication in Life Science Alliance. They further pointed out that some of your results need to be more carefully discussed, and that it would be good to include further quantifications and more insight into hexamer formation / into the stoichiometry of the observed complex.

We think that these issues can be addressed in a minor revision and by providing a detailed point-by-point response to the concerns raised. We would therefore like to invite you to revise your work for publication in Life Science Alliance. Please get in touch in case you would like to discuss individual revision points further while your work progresses.

Thank you for this interesting contribution to Life Science Alliance. We are looking forward to receiving your revised manuscript.

Reviewer reports from previous peer-review elsewhere:

Referee #1:

General summary and opinion

The authors use *in vitro* experiments to investigate the important functions of the histone chaperone FACT in both nucleosome assembly and nucleosome disassembly. The questions raised by the authors are relevant in the field, as it is currently not known in complete detail how the different activities of this essential histone chaperone FACT are regulated and how FACT recognizes the entire nucleosome. Using EMSAs and analytical centrifugation assays as the basis for their experiments, the authors show that FACT neither binds fully assembled nucleosomes, nor does it disassembles them, but rather FACT deposits a histone H2A-H2B dimer on the tetrasome. This results in a stable intermediate complex consisting of the chaperone FACT, the histone hexamer and DNA. The authors also show that longer DNA fragments can at least *in vitro* compete FACT off the H2A-H2B dimer, thus promoting hexasome and nucleosome formation. Although the observations presented in the manuscript are interesting, we feel that additional biophysical and cellular characterization would be necessary to validate some of the biophysical observations through independent assays, and to reveal the existence of provocative hexasomes *in vivo*. Overall, the findings to this referee appear too preliminary for publication.

Specific major concerns

The authors base their conclusions using two techniques - EMSA and ultracentrifugation. As currently conducted, the EMSA experiments are not very quantitative. However, since various components in the assays are labeled, it should be possible to quantify the results in the gels.

The novelty of some of the results is not sufficiently clear.

For example,

- Figure 2: it was already shown that FACT does not bind intact nucleosomes in EMSA experiments, as correctly cited by the authors in the Discussion (Tsunaka et al., 2016). This figure can probably be moved to Supplemental Information.

- Figure 7 - it was already shown that FACT stimulates transcription in vitro on a chromatinized template (Pavri et al. 2006). The added value of the current experiments in the manuscript are not sufficiently discussed in the context of the published state-of-the-art.

- In addition, based on the available crystal structures, it had already been suggested earlier and shown earlier that DNA and FACT binding to H2A-H2B are incompatible (for example in Kemble et al., 2015, or Hsieh et al. 2013, so that either DNA or FACT can bind to H2A-H2B). It is therefore not surprising that DNA would compete FACT off the H2A-H2B dimer. These papers should be discussed within the Discussion of a revised manuscript.

The intermediate complex consisting of a hexasome and FACT is interesting, but ought to be experimentally better characterized. Apart from Flag-IP and Flag peptide elutions, there is no direct evidence that this band contains the histone chaperone FACT. Other methods, such as gel filtration analysis, could be used to show that this intermediate contains FACT, histones and DNA.

We are also concerned whether the level of evidence for the existence of the hexasome is sufficiently robust. A better quantification method would be required to ascertain the stated stoichiometry. Native mass spectrometry may represent on such experimental avenue to substantiate their conclusions through an independent assay.

It would be important to know how the FACT tethers different hexasome components. Which domains of FACT are critically involved?

The presented model suggests that upon FACT loss, H2A-H2B dimer should be lost from nucleosomes in vivo. In Rhee et al. (Cell, 2014) the (distinct) authors identified nucleosome asymmetry and subnucleosomal particles in vivo using ChIP-exo experiments. An in vivo analysis of ChIP-exo profiles in FACT TS mutants would thus strengthen the relevance of this study.

Referee #2:

General summary and opinion

The manuscript 'The histone chaperone FACT modulates nucleosome structure by tethering its components' by Wang et al. describes how the histone chaperone FACT interacts with histone substrate to modulate nucleosome assembly. Several potential models have been proposed but the precise mechanism is still unclear. Previous data indicate that the MID domain of FACT interacts with a histone H3-H4 tetramer. By doing so, FACT disrupts H2A-H2B docking onto H3-H4, DNA interaction and nucleosome assembly. Additional sequence elements from FACT interact with H2A-H2B leading to the model that FACT maintains H2A-H2B tethered to a partially disassembled nucleosome.

The authors perform a series of biochemical experiments including sedimentation velocity AUC and nucleosome assembly assays. In agreement with previous data, Wang et al. show that FACT interacts with both H3-H4 and H2A-H2B. Mixing of FACT with H3-H4 results in aggregation of the complex while further addition of H2A-H2B results in a soluble complex. This result is interpreted to mean that a H2A-H2B dimer facilitates the proper interaction of FACT with H3-H4, apparently through the docking domain of H2A and H4. This interpretation is confirmed using a H4 mutant Y98H which disrupts binding of H3-H4 to FACT-H2A-H2B. The authors conclude that FACT stabilizes the interaction between H3-H4 and H2A-H2B.

On a short DNA substrate, the authors describe a ternary complex containing tetrasomes, H2A-H2B and apparently FACT. The authors propose that, on extended DNA substrates, the DNA would compete off H2A-H2B from FACT thus resulting in hexasome and nucleosome assembly. The overall conclusions are summarized in Fig. 8 where the authors propose that FACT enables H2A-H2B deposition onto tetrasomes to form a hexasome intermediate.

In general, it is not clear that the manuscript passes the threshold that would be required for publication. Results from the sedimentation velocity Experiments in Fig. 1 and 2 are not definitive enough to allow rigorous conclusions on the stoichiometry of the various complexes being assessed. It is not clear that data shown in Fig. 3-6 reveal new sufficiently novel insights into the mechanism of action of FACT. More specific concerns are indicated below.

Specific major concerns:

1. Mixing of FACT with H3-H4 results in aggregation of the complex while further addition of H2A-H2B results in a soluble complex. This result is interpreted to mean that H2A-H2B dimer facilitates the proper interaction of FACT with H3-H4. It is not clear that such a conclusion is rigorously supported by the data, as any ligand binding event can help to 'chaperone' assembly/solubility of a protein complex.

2. Fig. 3A: Considering that FACT interacts with the DNA binding surface of H3-H4, is surprising that FACT would facilitate tetrasome assembly. Instead it should be competitive. Is it possible that the effect observed is simply due

to the fact that the soluble pool of H3-H4 tetramers is increased in the presence of FACT?

3. Fig. 3B: The authors observed increased hexasome and nucleosome formation upon addition of FACT suggesting that FACT promotes the first steps of nucleosome assembly. Would a similar effect not been observed if FACT simply maintains solubility of otherwise aggregating histones? Therefore, statements that FACT has 'moderate nucleosome assembly activity' through 'facilitating tetrasome assembly' or through 'aiding H2A-H2B deposition onto tetrasomes' are not entirely convincing.

4. Fig. 6B: As the intermediate complex is still obtained despite using a H4 (Y98H) and H3 (I51A) mutant, it is not fully convincing that the complex relies on the direct interaction between H3-H4 and H2A-H2B, as suggested by the authors.

5. It is not clear that the slower migrating complexes seen in Fig. 3D, Fig. 4A, B are due to a genuine intermediate complex between FACT-H2A-H2B, H3-H4 and DNA. Instead such species could simply arise due to non-specific interactions between exposed basic histone surfaces and DNA.

6. Fig. 8: The model is not convincing: Why do the authors depict FACT as a H2A-H2 chaperone which only acts on tetrasomes that are already deposited on DNA? The largest binding interface of FACT with histones is through the H3-H4 interface, suggesting that a major function of FACT is to regulate DNA association of H3-H4 tetramers, a fact that is completely omitted from the authors' model. Collectively, it seems that the current model is that FACT binds simultaneously to H3-H4 tetramers and H2A-H2B dimers that have transiently dissociated from DNA to maintain the histones in a soluble pool.

Minor concerns

Fig. 3C: The authors titrate FACT and state that the fluorescence intensity from nucleosomal H4 increases, while the intensity of tetrasomal H4 decreases. There are no apparent tetrasomes present in these experiments. Do they mean hexasomes? Again, as an effect is only seen at very high FACT concentrations, could this be simply due to maintenance of histone solubility?

Fig. 3D: It seems impossible to distinguish hexasomes (lane 2) from non-specifically bound H2A-H2B (lane 4). Therefore it also seems unreasonable to state that hexasomes are the main assembly products upon addition of FACT.

Referee #3:

Wang et al. present new biophysical characterization of histones and nucleosomes interacting with human FACT. They show that a stable complex of hFACT forms with (H3-H4)₂ and a single H2A-H2B dimer, and they provide strong evidence supporting a model in which DNA competes with FACT to convert such complexes to hexasomes. The results clarify some longstanding issues in this field and will have high impact. The demonstration that FACT can bind to intermediates along the nucleosome disassembly/assembly pathway will also be important for future work in several areas of chromatin research. Several presentation issues need to be addressed before the manuscript will be suitable for publication but the quality and importance of the results are both high.

1. The use of sedimentation velocity AUC is quite rigorous and the data are of high quality. However, the authors should point out that this method detects primarily stable complexes, whereas the fluorescence dequenching assay they used previously to detect interactions between FACT and nucleosomes would also detect transient interactions. The possibility that FACT forms both unstable complexes and stable ones, with only the latter being described here, should be discussed along with the other possible explanations for the difference between their current and previous conclusions.

2. The data shown in Figure 2 demonstrating that hFACT alone does not disassemble nucleosomes is convincing and consistent with previous work, but the authors should mention that both yFACT and hFACT have only been shown to destabilize nucleosome structure in the presence of added HMGB family DNA binding protein cofactors. Ideally, this would be addressed experimentally but that is not considered necessary for this publication as it is sufficiently important to show that hFACT itself does not have this activity. But the result would have higher impact if it could be compared with other work showing the important role of DNA binding factors in collaboration with FACT.

3. Figure S3 shows quantitation of Fig 3C, but a similar analysis should be provided for 3A and other figures from which quantitative conclusions are drawn.

4. Two distinct migrating forms are described in Fig 4B, with their changes in distribution discussed in the text. Do the authors have any ideas about what the two forms denote? If so, this should be discussed, and if not this should be mentioned as the text is currently confusing.

5. The discussion of Fig 5A concludes that "FACT does not bind to the (H3-H4)₂ tetrasome in the absence of H2A-H2B dimer" but a second band at higher migration is seen in lane 3. Do the authors have any idea what this might be?

6. The conclusion from Fig 6B is not compelling. The amount of complex formed appears similarly reduced with both histone mutants, but this is described as normal in one case and reduced in the other. This is an important point as it addresses whether H2A-H2B interacts with H3-H4 in the complexes, so it should be quantitatively assessed to establish the strength of the conclusion.

7. Fig 7 has several issues to address, and the authors may want to consider whether or not these data add significantly enough to the manuscript to warrant their inclusion. First, the length of time of the assay should be reported here and discussed. If a 15 minute time point is used as in the Kuryan paper referenced, the amount of full length product represents an amount of read through that does not seem physiologically significant. This is an issue for this field, not just for this manuscript, but should be explicitly stated and discussed. Second, the discussion in the text implies that Kuryan et al. found a similar moderate effect of FACT on elongation, whereas those authors found no effect of FACT, only the RSC + Nap1 effect also seen here. Third, the level of production of full-length product should be quantified; by eye the last two lanes seem to be missing total signal (not the reduced FL production that is implied in the text) as there are no pauses seen but the FL form is less intense than in the previous lane. Does this mean there was less read-through or loss of product during handling? Finally, the authors state that the 1st and 2nd pause sites are not observed with RSC alone, but bands are visible on the gel in about the same locations until RSC is mixed with high levels of FACT. These pauses seem less distinct but they seem to be in the same sites as those observed with FACT. The important result of the experiment appears to be that FACT promotes some entry of RNA Pol II into nucleosomes (the low MW bands disappear, although this is not discussed), but enhances pausing at interpretable sites internal to the nucleosomes, and modestly improves FL transcript production, unlike what Kuryan et al reported. These pause sites seem to be common to the nucleosome barrier but are less prominent with RSC. Perhaps the discussion of the experiment can be focused on these points, as the current description is confusing and doesn't match the results presented well. Overall, the authors should consider what this experiment adds to the manuscript, state that clearly, and provide quantitative support for that conclusion.

Minor issues:

8. The manuscript requires additional editing for grammatical and typographical errors, and consistent use of the past tense when describing results presented here throughout.

9. In the introduction, the Jamai and Voth references in this passage should be switched:
"ChIP experiments in yeast suggest that yFACT also reassembles nucleosomes in the wake of RNA polymerase II (Jamai et al., 2009, Nguyen et al., 2013). Incorporation of new H3 in yeast gene bodies increases in the absence of Spt16 (Voth et al., 2014), suggesting that FACT contributes to the maintenance of pre-existing tetrasomes."

10. The discussion of the use of the "DMH3" experiment in Fig 1B is incomplete; the authors should draw the conclusion that tetramers are the binding partners explicitly.

11. The text discussing Fig 3D mentions asterisks denoting intermediate complexes; are arrows meant? No asterisks are visible in the figure. Also, consistent use of "FLAG" as the epitope should be adopted throughout.

12. The truncated Spt16 is called 1-932 on page 14, but 1-934 in the methods section.

1st Revision – authors' response

26 June 2018

Referee #1:

Specific major concerns

The authors base their conclusions using two techniques - EMSA and ultracentrifugation. As currently conducted, the EMSA experiments are not very quantitative. However, since various components in the assays are labeled, it should be possible to quantify the results in the gels. Gel quantification are notoriously imprecise and we thus feel it is better, in general, to let the results stand as qualitative findings.

The novelty of some of the results is not sufficiently clear.

For example,

- Figure 2: it was already shown that FACT does not bind intact nucleosomes in EMSA experiments, as correctly cited by the authors in the Discussion (Tsunaka et al., 2016). This figure can probably be moved to Supplemental Information.

We have used AUC to confirm that FACT does not bind to or disassemble nucleosome. This is the most rigorous in-solution assay based on first principles. We have therefore decided to leave this figure in its current position.

- Figure 7 - it was already shown that FACT stimulates transcription in vitro on a chromatinized template (Pavri et al. 2006). The added value of the current experiments in the manuscript are not sufficiently discussed in the context of the published state-of-the-art.

See response to reviewer 3. Our system is more simplified compared to Pavri et al. 2006, which contains many additional components that might regulate FACT (PARP1, PAF complex, etc).

- In addition, based on the available crystal structures, it had already been suggested earlier and shown earlier that DNA and FACT binding to H2A-H2B are incompatible (for example in Kemble et al., 2015, or Hsieh et al. 2013, so that either DNA or FACT can bind to H2A-H2B). It is therefore not surprising that DNA would compete FACT off the H2A-H2B dimer. These papers should be discussed within the Discussion of a revised manuscript.

Even if DNA and FACT binding to H2A-H2B are completely incompatible, this does not necessarily mean that DNA should be able to compete FACT off in a ternary complex. It is also possible that FACT can compete DNA off. Our competition assay clearly shows that DNA displaces FACT to form a hexasome and ultimately nucleosome. According to the reviewer's suggestion, we have included a better discussion of the previous literature (page 14).

"This is consistent with the published FACT-H2A-H2B crystal structure which shows that FACT-H2A-H2B interaction is incompatible with DNA-H2A-H2B interaction (Hsieh, Kulaeva et al., 2013, Kemble, McCullough et al., 2015).

-The intermediate complex consisting of a hexasome and FACT is interesting, but ought to be experimentally better characterized. Apart from Flag-IP and Flag peptide elutions, there is no direct evidence that this band contains the histone chaperone FACT. Other methods, such as gel filtration analysis, could be used to show that this intermediate contains FACT, histones and DNA.

We showed that the intermediate complex forms only in the presence of FACT, histones, and DNA. The presence of H2B and H3, and DNA was confirmed in the supershifted complex. M2 beads specifically recognize the FLAG tag, and FACT is the only component with a FLAG tag. The M2 pulldown assay specifically pulls down the intermediate complex. This is rigorous evidence to show that this intermediate complex contains histone chaperone FACT. To avoid any confusion, we added clarification on page 12.

"Nucleosomes (yellow), sub-nucleosomal complexes (green and red), and free DNA were enriched in the flow through (lane 3), while only the intermediate complexes containing FLAG-tagged FACT are specifically pulled down with the FLAG peptide. This complex also contains DNA, H2B, and H4 (yellow, lane 4)."

-We are also concerned whether the level of evidence for the existence of the hexasome is sufficiently robust. A better quantification method would be required to ascertain the stated stoichiometry. Native mass spectrometry may represent on such experimental avenue to substantiate their conclusions through an independent assay.

-It would be important to know how the FACT tethers different hexasome components. Which domains of FACT are critically involved?

-The presented model suggests that upon FACT loss, H2A-H2B dimer should be lost from nucleosomes in vivo. In Rhee et al. (Cell, 2014) the (distinct) authors identified nucleosome asymmetry and subnucleosomal particles in vivo using CHIP-exo experiments. An in vivo analysis of CHIP-exo profiles in FACT TS mutants would thus strengthen the relevance of this study.

The above three points are all good suggestions, but we believe are outside of the scope of this work.

Referee #2:

Specific major concerns:

1. Mixing of FACT with H3-H4 results in aggregation of the complex while further addition of H2A-H2B results in a soluble complex. This result is interpreted to mean that H2A-H2B dimer facilitates the proper interaction of FACT with H3-H4. It is not clear that such a conclusion is rigorously supported by the data, as any ligand binding event can help to 'chaperone' assembly/solubility of a protein complex.

We are not sure we understand this reviewer's point. We are indeed showing that H2A-H2B binding to FACT helps to chaperone assembly / solubility of the protein complex.

2. Fig. 3A: Considering that FACT interacts with the DNA binding surface of H3-H4, is surprising that FACT would facilitate tetrasome assembly. Instead it should be competitive. Is it possible that the effect observed is simply due to the fact that the soluble pool of H3-H4 tetramers is increased in the presence of FACT?

Many chaperones act like partially occluding the DNA-interacting regions on histones. We are not sure what the reviewer means by 'competitive', as the mechanism of histone deposition onto DNA by FACT could indeed be 'competitive' (i.e. the DNA binds better than FACT). At any rate, the 'tetrasome assembly activity mediated by FACT is weak at best, as pointed out in the text and further clarified throughout the text. We added clarification on page 9.

"To this end, an established in vitro nucleosome assembly assay was employed (Mattioli et al., 2017)."

3. Fig. 3B: The authors observed increased hexasome and nucleosome formation upon addition of FACT suggesting that FACT promotes the first steps of nucleosome assembly. Would a similar effect not be observed if FACT simply maintains solubility of otherwise aggregating histones? Therefore, statements that FACT has 'moderate nucleosome assembly activity' through 'facilitating tetrasome assembly' or through 'aiding H2A-H2B deposition onto tetrasomes' are not entirely convincing.

Overall, we changed the text to better reflect the weak tetrasome assembly activity of FACT. We believe the main activity of FACT is to tether the H2A-H2B dimer to the tetrasome.

4. Fig. 6B: As the intermediate complex is still obtained despite using a H4 (Y98H) and H3 (I51A) mutant, it is not fully convincing that the complex relies on the direct interaction between H3-H4 and H2A-H2B, as suggested by the authors.

We stated that the direct interaction between H3-H4 and H2A-H2B as "involved", not "rely on". Further clarification has been added to the main text at page 15.

'This indicates that H2A-H2B also interacts with H3-H4 in the intermediate complex, although the interactions are not as critical, as in in the histone hexamer-FACT complex (Figure 1D). This is consistent with our previous findings that H3I51A and H4Y98H cannot be refolded into histone octamers in the absence of DNA, but still assemble into (H3-H4)₂ nucleosomes in the presence of DNA in vitro (Ferreira, Somers et al., 2007; Hsieh et al., 2013; Ramachandran, Vogel et al., 2011). This indicates a role for DNA in tethering H3-H4 onto the FACT•(H2A-H2B) complex.'

5. It is not clear that the slower migrating complexes seen in Fig. 3D, Fig. 4A, B are due to a genuine intermediate complex between FACT-H2A-H2B, H3-H4 and DNA. Instead such species could simply arise due to non-specific interactions between exposed basic histone surfaces and DNA.

We showed that the intermediate complex forms only in the presence of FACT with histones and DNA. In addition, M2 beads specifically recognize FLAG tag, and in our intermediate complex, FACT is the only component with a FLAG tag. The M2 pulldown assay demonstrates that this intermediate complex containing FLAG-tagged FACT is specifically pulled down. This is solid evidence to show that this intermediate complex contains histone chaperone FACT. To avoid any confusion, we added clarification on Page 12.

"Nucleosomes (yellow), sub-nucleosomal complexes (green and red), and free DNA were enriched in the flow through (lane 3), while only the intermediate complexes with FLAG-tagged FACT is specifically pulled down with the FLAG peptide, which also contains DNA, H2B, and H4 (yellow, lane 4)."

6. Fig. 8: The model is not convincing: Why do the authors depict FACT as a H2A-H2 chaperone which only acts on tetrasomes that are already deposited on DNA? The largest binding interface of FACT with histones is through the H3-H4 interface, suggesting that a major function of FACT is to regulate DNA association of H3-H4 tetramers, a fact that is completely omitted from the authors' model. Collectively, it seems that the current model is that FACT binds simultaneously to H3-H4 tetramers and H2A-H2B dimers that have transiently dissociated from DNA to maintain the histones in a soluble pool.

FACT is a chaperone for both H2A-H2B and H3-H4. Even if the largest 'known' binding interface of FACT might be through the H3-H4 interface, there are several papers showing that FACT also binds to H2A- H2B.

As we pointed out in our manuscript, FACT tetrasome assembly activity is weak at best. This does not contradict the finding that FACT interacts with H3-H4; indeed, many chaperones bind histones yet have only very weak assembly function. In our manuscript we have identified a function of FACT that could not have been simply inferred from its binding preferences to individual histone components.

Minor concerns

Fig. 3C: The authors titrate FACT and state that the fluorescence intensity from nucleosomal H4 increases, while the intensity of tetrasomal H4 decreases. There are no apparent tetrasomes present in these experiments. Do they mean hexasomes? Again, as an effect is only seen at very high FACT concentrations, could this be simply due to maintenance of histone solubility?

Thank you for pointing this out - we fixed this error. Please see the texts at page 10: *"hexasomal H4 decreases (Figure 3C; quantification in S3A)."*

Fig. 3D: It seems impossible to distinguish hexasomes (lane 2) from non-specifically bound H2A-H2B (lane 4). Therefore it also seems unreasonable to state that hexasomes are the main assembly products upon addition of FACT.

In the main text we clarified that in Lane 4 there is no hexasome formed. Instead, it indeed represents histones that are bound to DNA nonspecifically. Lane 6-10 actually shows hexasome formation, and the hexasome band migrates faster in the gel in lane 6-10.

Referee #3:

Wang et al. present new biophysical characterization of histones and nucleosomes interacting with human FACT. They show that a stable complex of hFACT forms with (H3-H4)₂ and a single H2A-H2B dimer, and they provide strong evidence supporting a model in which DNA competes with FACT to convert such complexes to hexasomes. The results clarify some longstanding issues in this field and will have high impact. The demonstration that FACT can bind to intermediates along the nucleosome disassembly/assembly pathway will also be important for future work in several areas of chromatin research. Several presentation issues need to be addressed before the manuscript will be suitable for publication but the quality and importance of the results are both high.

1. The use of sedimentation velocity AUC is quite rigorous and the data are of high quality. However, the authors should point out that this method detects primarily stable complexes, whereas the fluorescence dequenching assay they used previously to detect interactions between FACT and nucleosomes would also detect transient interactions. The possibility that FACT forms both unstable complexes and stable ones, with only the latter being described here, should be discussed along with the other possible explanations for the difference between their current and previous conclusions. Compared to fluorescence dequenching assay, AUC is a better way to define complex as it relies on first principles only. We believe that FRET, when done under equilibrium conditions, would not detect transient interactions, but it is prone to artefacts from other events that can result in fluorescence dequenching.

2. The data shown in Figure 2 demonstrating that hFACT alone does not disassemble nucleosomes is convincing and consistent with previous work, but the authors should mention that both yFACT and hFACT have only been shown to destabilize nucleosome structure in the presence of added HMGB family DNA binding protein cofactors. Ideally, this would be addressed experimentally but that is not considered necessary for this publication as it is sufficiently important to show that hFACT itself does not have this activity. But the result would have higher impact if it could be compared with other work showing the important role of DNA binding factors in collaboration with FACT.

Thank you for your suggestion. We added the discussion of the importance of HMG domain at page 17.

'In addition, the presence of HMGB DNA binding protein cofactors is required for FACT to reorganize nucleosomes (McCullough, 2018). Overall, the consensus is that FACT per se lacks the ability to modulate nucleosome.'

3. Figure S3 shows quantitation of Fig 3C, but a similar analysis should be provided for 3A and other figures from which quantitative conclusions are drawn.

Since we believe that the tetrasome assembly effects of FACT (shown in Fig. 3A) are weak, we decided against quantification and instead changed the text to de-emphasize FACT tetrasome assembly activity.

4. Two distinct migrating forms are described in Fig 4B, with their changes in distribution discussed in the text. Do the authors have any ideas about what the two forms denote? If so, this should be discussed, and if not this should be mentioned as the text is currently confusing.

Thank you for your suggestions. In Fig.4C, when more H2A-H2B is added, the lower band shifted to the upper band. We added clarification on page 12:

'This indicates that the slower migrating species contains one more H2A-H2B than the faster migrating species.'

5. The discussion of Fig 5A concludes that "FACT does not bind to the (H3-H4)₂ tetrasome in the absence of H2A-H2B dimer" but a second band at higher migration is seen in lane 3. Do the authors have any idea what this might be?

We do not have solid data to determine the higher migration band. Based on our knowledge, the higher migration band may be tetrasome in a different position on the DNA, or perhaps with different stoichiometry of histones.

6. The conclusion from Fig 6B is not compelling. The amount of complex formed appears similarly reduced with both histone mutants, but this is described as normal in one case and reduced in the other. This is an important point as it addresses whether H2A-H2B interacts with H3-H4 in the complexes, so it should be quantitatively assessed to establish the strength of the conclusion. We agree with this reviewer. This result also should be seen in context with the data in Fig.1D. Even though the interface is required for the formation of the intermediate complex, its importance is downgraded in the presence of the stabilizing effects of DNA.

We stated the direct interaction between H3-H4 and H2A-H2B as "involved", not "rely on". We have included clarification in the main text at page 15.

"This indicates that H2A-H2B also interacts with H3-H4 in the intermediate complex, although not critical, as it does in the histone hexamer-FACT complex (Figure 1D). This corresponds to the fact that H3151A and H4Y98H cannot be refolded into histone octamers in the absence of DNA, but still assemble into (H3-H4)₂ nucleosomes in the presence of DNA in vitro (Ferreira, Somers et al., 2007; Hsieh et al., 2013; Ramachandran, Vogel et al., 2011). Therefore, this indicates an additional role of DNA in tethering the H3-H4 onto FACT•(H2A-H2B) complex."

7. Fig 7 has several issues to address, and the authors may want to consider whether or not these data add significantly enough to the manuscript to warrant their inclusion. First, the length of time of the assay should be reported here and discussed. If a 15minute time point is used as in the Kuryan paper referenced, the amount of full length product represents an amount of read through that does not seem physiologically significant. This is an issue for this field, not just for this manuscript, but should be explicitly stated and discussed. Second, the discussion in the text implies that Kuryan et al. found a similar moderate effect of FACT on elongation, whereas those authors found no effect of FACT, only the

RSC + Nap1 effect also seen here. Third, the level of production of full-length product should be quantified; by eye the last two lanes seem to be missing total signal (not the reduced FL production that is implied in the text) as there are no pauses seen but the FL form is less intense than in the previous lane. Does this mean there was less read-through or loss of product during handling? Finally, the authors state that the 1st and 2nd pause sites are not observed with RSC alone, but bands are visible on the gel in about the same locations until RSC is mixed with high levels of FACT.

These pauses seem less distinct but they seem to be in the same sites as those observed with FACT. **The important result of the experiment appears to be that FACT promotes some entry of RNA Pol II into nucleosomes (the low MW bands disappear, although this is not discussed), but enhances pausing at interpretable sites internal to the nucleosomes, and modestly improves FL transcript production, unlike what Kuryan et al reported.** These pause sites seem to be common to the nucleosome barrier but are less prominent with RSC. Perhaps the discussion of the experiment can be focused on these points, as the current description is confusing and doesn't match the results presented well. Overall, the authors should consider what this experiment adds to the manuscript, state that clearly, and provide quantitative support for that conclusion.

We thank the reviewer for his/her suggestions and for constructive advice in interpreting our results. We have changed the entire paragraph discussing these results according to this reviewer's suggestions (page 15-16, last paragraph before the discussion).

Minor issues:

8. The manuscript requires additional editing for grammatical and typographical errors, and consistent use of the past tense when describing results presented here throughout.

9. In the introduction, the Jamai and Voth references in this passage should be switched:

"ChIP experiments in yeast suggest that yFACT also reassembles nucleosomes in the wake of RNA polymerase II (Jamai et al., 2009, Nguyen et al., 2013). Incorporation of new H3 in yeast gene bodies increases in the absence of Spt16 (Voth et al., 2014), suggesting that FACT contributes to the maintenance of pre-existing tetrasomes."

Thank you for this correction. This error has been fixed (bottom of page 4).

10. The discussion of the use of the "DMH3" experiment in Fig 1B is incomplete; the authors should draw the conclusion that tetramers are the binding partners explicitly.

Thank you. We have added the text on page 7:

' *This indicates that H3-H4 exists as a (H3-H4)₂ tetramer in the complex with FACT and H2A-H2B* '

11. The text discussing Fig 3D mentions asterisks denoting intermediate complexes; are arrows meant? No asterisks are visible in the figure. Also, consistent use of "FLAG" as the epitope should be adopted throughout.

Thank you, we fixed these errors (See text on page 10; and page 11)

12. The truncated Spt16 is called 1-932 on page 14, but 1-934 in the methods section.

Thank you for carefully reviewing our manuscript. This error has been fixed. See texts at page 14.

"FACT Δ CTD (Spt161-934), was previously shown to be deficient for H2A-H2B binding (Winkler, Muthurajan et al., 2011)."

2nd Editorial Decision

27 June 2018

Thank you for submitting your revised manuscript entitled "The histone chaperone FACT modulates nucleosome structure by tethering its components". We appreciate your response to the reviewer comments and the changes introduced in the manuscript, and we are happy to publish your paper in Life Science Alliance pending final revisions necessary to meet our formatting guidelines.
